# Visualization of Nuclease- and Serum-Mediated Chromatin Degradation with DNA–Histone Mesostructures

**DOI:** 10.3390/ijms24043222

**Published:** 2023-02-06

**Authors:** Midori L. Wasielewski, Katherine Nguyen, Srilakshmi Yalavarthi, Pallavi Ekbote, Priyan D. Weerappuli, Jason S. Knight, Shuichi Takayama

**Affiliations:** 1Wallace H. Coulter Department of Biomedical Engineering, Georgia Institute of Technology and Emory University, Atlanta, GA 30332, USA; 2The Parker H. Petit Institute for Bioengineering and Bioscience, Georgia Institute of Technology, Atlanta, GA 30332, USA; 3Division of Rheumatology, Department of Internal Medicine, University of Michigan, Ann Arbor, MI 48109, USA; 4Biointerfaces Institute, University of Michigan, Ann Arbor, MI 48105, USA

**Keywords:** neutrophil extracellular traps (NETs), chromatin degradation, NETs in autoimmunity

## Abstract

This study analyzed the nuclease- and serum-driven degradation of millimeter-scale, circular DNA–histone mesostructures (DHMs). DHMs are bioengineered chromatin meshes of defined DNA and histone compositions designed as minimal mimetics of physiological extracellular chromatin structures, such as neutrophil extracellular traps (NETs). Taking advantage of the defined circular shape of the DHMs, an automated time-lapse imaging and image analysis method was developed and used to track DHM degradation and shape changes over time. DHMs were degraded well by 10 U/mL concentrations of deoxyribonuclease I (DNase I) but not by the same level of micrococcal nuclease (MNase), whereas NETs were degraded well by both nucleases. These comparative observations suggest that DHMs have a less accessible chromatin structure compared to NETs. DHMs were degraded by normal human serum, although at a slower rate than NETs. Interestingly, time-lapse images of DHMs revealed qualitative differences in the serum-mediated degradation process compared to that mediated by DNase I. Importantly, despite their reduced susceptibility to degradation and compositional simplicity, the DHMs mimicked NETs in being degraded to a greater extent by normal donor serum compared to serum from a lupus patient with high disease activity. These methods and insights are envisioned to guide the future development and expanded use of DHMs, beyond the previously reported antibacterial and immunostimulatory analyses, to extracellular chromatin-related pathophysiological and diagnostic studies.

## 1. Introduction

Extracellular chromatin, such as neutrophil extracellular traps (NETs), although beneficial in fighting bacteria [1,2], can also have detrimental effects such as prolonging tissue repair, promoting unwanted coagulation, and serving as a nidus for autoantigen generation [2,3,4]. Tools and methods to conveniently evaluate the ability of bodily fluids to degrade extracellular chromatin, therefore, are sought after for their potential to understand and discern disease states [5,6,7]. In systemic lupus erythematosus (SLE), for example, decreased NET degradation by serum may indicate imminent flares, including serious manifestations such as lupus nephritis [8]. With an eye towards bioengineering useful tools for the evaluation of extracellular chromatin degradation by bodily fluids and in vitro cell cultures, this study describes a time-lapse imaging and image processing method for use in the analysis of the nuclease- and serum-mediated degradation of a NET-mimetic material that our lab recently developed with a defined composition and mesoscale structures.

DNA–histone mesostructures (DHMs) [9] and microwebs [10,11] resemble aspects of the chromatin backbone structure of NETs. For example, we previously reported that deoxyribonuclease I (DNase I) degrades both DHMs [9] and microwebs [10] in a manner that resembles the nuclease-mediated degradation that NETs undergo. Here, we analyzed DHM degradation in more detail including a comparison to NETs. The types of nucleases tested were also expanded to include micrococcal nuclease (MNase), which has a different specificity from DNase I. MNase is known to preferentially cut the linker DNA between nucleosomes well before it cleaves DNA that is wrapped around octamers [12,13]. DNase cleaves DNA less selectively with regards to DNA strands wrapped around histones. We also performed an exploratory analysis of DHM degradation by human serum. A revised method of DHM preparation that provides structures that are more consistent and wash resistant was also developed for more reliable degradation analyses. Finally, the degradation process was analyzed in more detail by visualizing DHMs with time-lapse imaging and automated analysis of the many images captured. DHMs spotted into 96-well microplates have a defined initial mesoscale (sub-millimeter scale) structure that facilitates image-based degradation monitoring (Appendix A) in ways not possible with the numerous smaller structures presented by NETs. The time-lapse imaging of DHM degradation by serum revealed a process involving a combination of structural dissolution and detachment not observed in DNase-I-mediated degradation. Interestingly, DHMs, unlike NETs, were not degraded well by MNase. Compared to NETs, we also found that DHMs degraded slower when exposed to serum. Despite these differences, DHMs could distinguish differences in the ability of normal control vs. severely-ill SLE patient serum in order to degrade them. 

## 2. Results

### 2.1. DNA–Histone Mesostructures (DHMs) Can Be Engineered into Consistent Chromatin Structures for Degradation Experiments

DHM preparation procedures were modified from our previous report [9] for consistency and wash resistance (Figure 1A). In this improved method, each microwell was poly-L-lysine pre-treated to make the DHMs resistant to delamination during the initial wash steps of an experiment. Adding a rinse step with a bovine serum albumin (BSA) solution was found to further enhance consistency. Neutrophil-derived NETs were formed on coverslips, fixed, and immunostained with neutrophil elastase for visual comparison (Figure 1B). Additional images of NETs stained with SYTOX Green were visualized in Appendix A. To begin a degradation experiment, DHMs were rehydrated from a dried state and stained with the fluorescent label SYTOX Green as depicted in Figure 1C. For comparison, the protocol required to generate and stain NETs from neutrophils for a NET degradation assay is outlined in Figure 1D.

### 2.2. Image Capture and Analysis for Tracking DHM Degradation

Each DHM-containing microwell was monitored by time-lapse imaging using an in-incubator microscope (Incucyte^®^ S3, Essen BioScience, Ann Arbor, MI, USA) (Figure 2A). Figure 2B shows an example of a DHM degradation time series upon incubation with 10 U/mL of DNase I. Under these conditions, the raw green fluorescence unit (GFU) readings captured both the DHM structure remaining as well as the fluorescence of degradation products in the solution. This resulted in a reading of 69.8% residual GFUs at 12 h despite the DHM structure being completely degraded upon human visual inspection (Figure 2B, top row). For better automated discernment of DHM signal from degradation product fluorescence, an image segmentation algorithm was utilized to segment the DHM shape and area. First, a Top-Hat method for background correction was applied to threshold images, which created binary images (Figure 2B, center row). A machine-learning algorithm (Incucyte^®^ Basic Analyzer 2022A, Essen BioScience, Ann Arbor, MI, USA) was then used to produce an object mask of the localized DHM signal. The resulting DHM area mask, pictured in magenta and with units μm^2^ per image, tracked structural changes in response to degradation treatment, showing 0.06% residual DHMs at 12 h (Figure 2B, bottom row). The readouts from the two automated image analysis and quantification methods in Figure 2B are plotted in Figure 2C, with the inferior whole-well fluorescence signal (green) analysis method compared to the superior DHM area (magenta) method. This DHM-area-based image processing method also worked well with 5% serum (Appendix A) where there was a relatively high background signal from the beginning. 

### 2.3. DNase I Degrades DHMs in a Dose-Dependent Manner

We analyzed DHM degradation using a three-fold dilution series of DNase I and image processing as described above (Figure 3A). Total DNase digestion was observed with the highest DNase I concentration of 10 U/mL by 12 h. Lower DNase I concentrations did not completely degrade the DHMs within the 48 h analysis period. Nuclease buffer in the absence of DNase I served as a negative control condition. The residual DHM area was calculated using the final DHM area as a percentage of each structure’s corresponding initial baseline DHM area with the equation below:(1)$$\%\;Residual=\frac{Final_{Area}}{Initial_{Area}}*100\%.$$

The mean for each treatment is plotted as *%Residual* in Figure 3B. The total extent of DHM degradation increased in a dose-dependent fashion when digested with DNase I. The residual DHM structures after nuclease digestion in plots (A) and (B) are visualized in Figure 3C. The fluorescent, green channel images revealed the partial digestion of low nuclease concentrations with small amounts of edge delamination and the center dimming with increasing DNase I concentration. Residual structures were very faint under 3.33 U/mL DNase conditions and degraded fully after 10 U/mL DNase incubation.

### 2.4. DHM Degradation by Human Serum Samples

Next, we applied the assay to characterize DHM degradation in response to representative human serum in vitro by testing a two-fold dilution series of commercially available human serum (H4522; Sigma-Aldrich, St. Louis, MO, USA) to serve as a healthy sample with normal nuclease activity. Nuclease buffer in the absence of serum served as a negative control condition. The DHM area in each well was measured and calculated as described above. Similar to DNase I, a dose-dependent response was observed. DHMs were degraded faster and to a greater extent in response to higher serum concentrations (Figure 4A). The residual DHM area was calculated using Equation 1, and the mean for each treatment was plotted as %Residual in Figure 4B. Representative images of the residual DHMs after the 48-h incubation (Figure 4C) show serum dose dependence, and, compared to DNase I treatment, relatively more detachment of still relatively intact mesh structures. This leads to DHM structures that detach from the edges and fold up onto themselves (for example, Figure 4C 1.25%). The delamination effect of serum on DHMs is attributed, at least in part, to digestion by calcium- and magnesium-dependent nucleases in the serum (Appendix A), as DNases in serum require calcium and magnesium to digest DNA. The serum treatment was not able to degrade DHMs in the absence of calcium and magnesium cations in the nuclease buffer. Additionally, serum did not degrade DHMs in the presence of EDTA, which chelates calcium and magnesium ions (Appendix A). In addition to providing a defined composition and a storage-stable chromatin mesh, DHMs have a defined sub-millimeter-scale structure that changes its overall shape in different ways during different degradation processes. 

### 2.5. DHMs Are More Resistant to Degradation Than NETs

DHM degradation was compared to NET degradation in vitro. DHMs and NETs were incubated with the following conditions: DNase I at 10 U/mL, MNase at 10 U/mL, 5% control human serum, or nuclease buffer control. These concentrations were selected as they are commonly used in NET degradation assays to allow nuclease activity comparisons in the samples. In serum, the 5% dilution additionally enabled optical clarity for real-time monitoring. DNase at 10 U/mL was selected as a positive control to fully degrade DHMs, which exceed physiological DNase I levels ranging roughly from 20 ng/mL [14,15] to 37 ng/mL [16] in the context of testing serum samples. The resulting DHM degradation is summarized in (Figure 5A). DHMs were minimally degraded under all conditions at 2 h but were degraded by serum and DNase I at 24 h, with an average 8.5% and 3.5% residual structures, respectively. DHMs were minimally degraded by MNase despite a 24 h incubation, with an average of 96.9% residual structures. For comparison, cell-derived NETs were formed in 96-well plates and their degradation was measured to determine %Residual NETs. NETs were degraded after a 1.5-h incubation with control serum having 26.3% residual structures, DNase having 3.8% residuals, and MNase having 2.2% residuals (Figure 5B). These comparisons revealed that DHMs degrade slowly when incubated with normal serum and nucleases relative to NETs. While NETs are degraded after a 1.5-h incubation, DHMs require a 24-h incubation for a comparable degradation extent. DHM residual structures were captured with the Incucyte S3^®^ (Essen BioScience, Ann Arbor, MI, USA) (Figure 5C, top row). The residual structure of serum-treated DHMs was a gathered aggregate at the center of the well. DNase treatments led to a more uniform dimming of the entire DHM structure. MNase, like nuclease buffer incubation, did not degrade the DHMs. NETs were imaged by immunofluorescence microscopy (Figure 5C, bottom row) to visualize representative images of their extent of degradation by the treatments described above. NETs were fixed and labeled with neutrophil elastase (in green), and neutrophil nuclei were co-stained with Hoechst stain (in blue). As expected, NETs incubated with DNase I, MNase I, and serum conditions were visibly degraded, showing decreased NETs compared to the nuclease buffer control condition. 

### 2.6. DHMs Show Similarities to NETs with Regards to Degradation by SLE Serum

We tested DHM degradation by serum samples from patients with systemic lupus erythematosus (SLE). DHMs pre-stained with 1 μM SYTOX Green were incubated with degradation treatments for 12 h, and residual images were acquired with fluorescent microscopy (Figure 6). Representative serum samples were selected to include normal controls, an SLE patient with low disease activity, and an SLE patient with high disease activity. The patient with high disease activity was known to have defective NET degradation based on pilot experiments that were part of our prior work [7]. These serum samples were diluted to 5% in nuclease buffer and incubated with DHMs. DHMs structures were efficiently degraded by 5% control serum and 5% SLE serum (low disease activity). In contrast, significant residual DHMs remained when incubated with serum from the SLE patient with high disease activity. Although DHMs have a minimal chromatin backbone composition and show slower degradation than NETs, they were able to mimic NETs in being degraded to a greater extent by serum from normal control donors than lupus donors. 

## 3. Discussion

Our group previously described a minimal NET-mimetic microweb, comprised of DNA and histones, for their ability to mimic aspects of the bacteria-suppressive features of NETs [10,11]. We have also described a surface-attached NET-mimetic biomaterial we call DHMs, which have a defined mesoscale structure also comprised of DNA and histones, for their NET-mimetic, immune-cell-activating features [9]. The microbicidal and cell-activating functions of these bioengineered NET-mimetic biomaterials, however, may be affected by their degradation rate. Therefore, in this study, we characterized the nuclease- and serum-mediated degradation of DHMs, with additional comparisons to NETs. Furthermore, given the reported potential diagnostic applications of NET degradation [5,6,7], we also evaluated whether DHM degradation rates differ between treatment with normal versus SLE serum.

The degradation assay was designed to quantify the kinetics and visualize the degradation, including detachment and associated shape changes, of DHM structures in real time. To select a time frame of observation, we considered the treatment incubation time of NET degradation assay protocols, which range widely from 90 min to 24 h [17]. Physiologically, it is unknown how long NETs persist in circulation or in tissues, with reports ranging from hours [18] to weeks [19]. We optimized the time frame to be one automated image capture per hour for at least 24 h, which allowed observations of the gradual degradation responses that DHMs exhibit, and the full extent of degradation for a given treatment once it reached a plateau. 

Several fluorescence-based NET degradation assays rely on end-point measurements taken before and after the degradation incubation in a buffer or PBS [7,20]. It is challenging to report live fluorescence kinetics because serum introduces a great deal of signal to the wells, which interferes with the pre-stained NET signal. Since our study involved the real-time monitoring of wells that contain serum, we developed an image segmentation method to enable the automated real-time quantification of DHM area change to overcome the background serum samples and confounding fluorescence signals from degraded DHM fragments in suspension. The defined starting shape of the DHMs coupled with the image segmentation method allowed the visualization of different degradation patterns from different treatments, as well as the quantitative tracking of DHM area. Area metrics, when paired with DHM raw intensity values, could also be useful in capturing the brightness of the structures during instances of delamination, where folding of the structure could lead to over- and underestimation of the area. There is an opportunity to improve the signal-to-noise ratio of DHMs in future work, for simplified real-time fluorescence tracking that does not rely on image analysis.

DNase I treatment led to gradual DHM fading consistent with the gradual digestion of the DNA backbone into smaller nucleotide fragments (Figure 3). Buffer controls caused minimal DHM shape change. Treatment with 5% normal serum led to more detachment in the early phases accompanied by a slower digestion process, leading to the formation of gathered aggregates during the intermediate stages of the degradation process (Figure 4). The differences in how DHM degradation proceeded between DNase I and serum are consistent with reports that there are mechanisms beyond DNase I that contribute to NET degradation in vivo [21]. We note that the use of an in-incubator microscope has previously been used in the real-time imaging of NETosis [22,23] but not NET degradation. Our DHM degradation analysis method expands the use of such instruments for NET-degradation-related studies. 

Compared to NETs, DHMs were degraded more slowly by diluted serum (Figure 5). Additionally, we observed minor variability between serum aliquots of the same origin, as observed between the DHM degradation of 5% serum in Figure 4A, which was slower to degrade than 5% serum in Figure 5A. To dissect the differences between the cell-based NETs and DHMs in the presence of nucleases, we tested both MNase and DNase. DNase I and MNase are two of the most commonly reported nucleases in NET degradation assays [17]. DNase I is active in human blood [16,24], while MNase is produced by bacteria; however, MNase is of interest because it is known to preferentially cut the linker DNA between nucleosomes well before it cuts DNA that is wrapped around octamers [12,13]. DNase cleaves DNA less selectively with regards to DNA strands wrapped around histones. DHMs and NETs were similarly fully degraded by 10 U/mL DNase. In particular, we were surprised that DHMs were not susceptible to degradation by 10 U/mL MNase, whereas NETs were readily degraded under the same conditions. During NETosis, histone citrullination mediates chromatin decondensation in neutrophils and granulocytes via peptidyl arginine deiminase 4 (PAD4) [25]. PAD4 facilitates NETosis, and it has been reported that MNase more readily digests externalized NETs after PAD4 treatment [25]. We speculate that these and other differences may result in DHM chromatin fibers being less accessible compared to those found in NETs. Other nucleases of interest to compare in future applications are the leukocyte-secreted nuclease DNase IL3 [26], and nucleases secreted by common Gram-positive bacteria to digest NETs as an evasion mechanism [27]. 

We note that we did not normalize the enzyme:DNA ratio between DHMs and NETs, which could impact the length of oligonucleotide byproducts and therefore the degradation accessibility. The coverage of DHMs was larger, and centered, versus NETs which were distributed across the well but individually in the micrometer size range. Despite the increased aggregation of DHMs, by our estimates, cell-derived NETs were composed of 6.7 times more DNA than DHMs per well. The DHMs were fabricated with a mass of 30 ng DNA per well, while the cellular NETs in this study had approximately 180 ng to 240 ng per well (equivalent to 30,000–40,000 NETotic cells per well of 100,000 plated neutrophils stimulated with a 20 nM phorbol 12-myristate 13-acetate (PMA) concentration). Despite the lower DNA amount per well, the DHMs still were slower to degrade using the same 10 U/mL MNase treatment, hence our conclusion that there are likely also structural differences that are behind the lower DHM degradation by MNase.

Impaired NET clearance contributes to the onset and progression of autoimmunity such as in SLE [2]. NETs become a source of self-antigens that can generate autoantibodies against components of NETs, such as anti-nuclear antibodies (ANAs). ANAs may lead to SLE flares by forming immune complexes that deposit on, and damage, skin and kidney tissues [4,8]. NET degradation assays have been reported for their potential to discern between disease states and predict SLE flares [28]. Here, we tested the potential of serum-mediated DHM degradation to also relate to SLE disease states. DHMs were degraded less by serum from a patient with higher disease activity (higher SLEDAI score) (Figure 6), consistent with the lowered chromatin and NET degradation observed in active SLE [4,20]. These results suggest promise for the further development of DHM degradation assays of clinical samples. The convenient-to-use protocol and stable long-term storage of DHMs, which is not possible with NETs, may facilitate translational research. On the other hand, our analysis showed that DHMs were more resistant to nuclease- and serum-mediated degradation compared to NETs, suggesting that modifications may be needed in how DHMs are prepared to better mimic the degradation profiles of NETs. Future development can also improve the well-to-well homogeneity of the DHM structures upon drying, which could introduce differences in DNA density. There is inherent biological variability in the nuclease and NET degradation activity among serum donors; therefore, these improvements can enable the application of the DHM degradation assay to a larger population of autoimmune patients in order to address this variability.

## 4. Materials and Methods

### 4.1. DNA–Histone Mesostructure (DHM) Fabrication

DHMs were fabricated by the step-wise spotting of DNA and histone droplets into wells of standard 96-well microplates. To increase the binding efficiency of DHMs, untreated 96-well microplates (CLS3370; Corning, Corning, NY, USA) were coated with 0.001% poly-L-lysine (P4707; Sigma-Aldrich, St. Louis, MO, USA) for 10 min and rinsed once with water. Once dry, the DNA layer was prepared by combining a solution of 0.1 mg/mL lambda-phage methylated DNA (D9768; Sigma-Aldrich, St. Louis, MO, USA) with a 400 mM trehalose solution (T-104-4; Pfanstiehl, Waukegan, IL, USA) at a 1:1 ratio. Suspending the soluble DNA within a trehalose phase added stability, localization, and uniformity to the final structures. The trehalose phase could be washed off prior to testing, leaving only the chromatin fibers in each well. As a note, histones could not be pre-mixed with the DNA solution as they instantly formed fibers and therefore could not be dispensed with liquid-handling techniques. 

A sub-microliter droplet, 0.6 μL, of the DNA–trehalose solution was dispensed into each well. The droplets were vitrified for 24 h in a vacuum desiccator. A 0.6 μL droplet containing a 0.5 mg/mL histone solution (H9250; Sigma-Aldrich, St. Louis, MO, USA) was then deposited over the vitrified DNA spot and dried for another 24 h in a vacuum desiccator. The concentration of DNA and histone solutions, as well as the volume, can be modified to vary the resulting morphology and compaction of the DHM chromatin structure. The binding of histones to DNA strands resulted in a defined sub-millimeter mesh of condensed chromatin fibers within the boundaries of the initial droplet area. Once fabricated, the resulting vitrified fibrous DNA structures were stable at room temperature until use. 

### 4.2. Degradation Assay Using DHMs

Rehydration and fluorescent staining of DHMs. Each DHM structure was rehydrated from its dry state, and simultaneously, the DNA component was fluorescently stained by adding 100 μL of a 1 μM SYTOX Green (S7020; Thermo Fisher Scientific, Waltham, MA, USA) in PBS for one hour, protected from light. The DHMs were washed three times in PBS to remove the trehalose phase and rinsed once with a 1% bovine serum albumin (BSA) solution (126609; EMD Millipore, Burlington, MA, USA) in PBS to further stabilize the structures. 

Sample preparation. DHMs were treated with DNase I from bovine pancreas (D4263; Sigma-Aldrich, St. Louis, MO, USA), MNase (88216; Sigma-Aldrich, St. Louis, MO, USA), or thawed normal control serum (H4522, or S1-M; reagents from Sigma-Aldrich, St. Louis, MO, USA) at the specified concentrations above. DNase I units (U) represented Kunitz units per manufacturer information. Serum was commercially available from pooled donor samples collected from the clots of healthy normal humans. With regards to H4522 male AB blood, per the manufacturer, male blood is chosen because female serum can have more antibodies resulting from pregnancy. Type AB serum is used as anti-A and/or anti-B antibodies are absent, which are found in other blood types. Serum samples were aliquoted upon receipt and stored at −80 °C until use. All degradation treatment samples were diluted in nuclease buffer (10 mM Tris–HCl pH 7.5, 10 mM MgCl_2_, 2 mM CaCl_2_, and 150 mM NaCl). Nuclease buffer solution served as the background control.

Real-time imaging procedure. To visualize DHM degradation, each well was gently aspirated, and 100 μL of the sample treatment was added to each DHM structure. The degradation was monitored continuously for 24 to 48 h at 37 °C and 5% CO_2_, using an Incucyte S3^®^ Live-Cell Analysis System (Essen BioScience, Ann Arbor, MI, USA). Each sample was tested at least in triplicate. After 5 min in the incubator, the lid and the bottom of the plate were wiped with a kimwipe to remove any condensation. Automated images were captured every hour using a 4X objective with brightfield and green (400-ms) channels. This enabled the real-time visualization of degradation for each DHM structure. After the degradation period, the residual DHMs were imaged using a 20X objective in brightfield and green fluorescence channels.

Degradation analysis. The instrument’s image analysis software, Incucyte^®^ Software 2022A (Essen BioScience, Ann Arbor, MI, USA) was used to quantify the kinetics and the extent of DHM degradation throughout the assay. The Incucyte^®^ Software Basic Analyzer algorithm is an image processing tool applied to define and segment a DHM object area for each image. First, images were selected to train the image segmentation algorithm to identify which objects to be classified as DHMs. The image thresholding and segmentation parameters were set to a radius of 100 μm, and a threshold value of 2 GCU, with Top-Hat background correction. The software generated previews of the DHM object mask using the selected object definition parameters to confirm that the green object mask captured the DHMs. Once the image segmentation definition was finalized, the algorithm processed all the captured images to generate a DHM mask area, observed as a magenta overlay in each image and calculated as μm^2^ per image by the Incucyte^®^ Software 2022A (Essen BioScience, Ann Arbor, MI, USA). The DHM mask area values were exported for analysis using GraphPad Prism 9 software (GraphPad Software, San Diego, CA, USA).

### 4.3. NET Degradation Assay

PMA-stimulated NETs were degraded as previously described with minor modifications [7]. Briefly, microplates were coated with 0.001% poly-L-lysine (P4707; Sigma-Aldrich, St. Louis, MO, USA) for 10 min and rinsed once with water and allowed to air dry. Purified control neutrophils were resuspended in RPMI media supplemented with L-glutamine (Gibco; Thermo Fisher Scientific, Waltham, MA, USA). Next, 1 × 10^5^ neutrophils were seeded into each well 96-well black-wall clear-bottom tissue culture plate (Thermo Fisher Scientific, Waltham, MA, USA). To induce NET formation, cells were incubated in the presence of 20 nM PMA (Sigma-Aldrich, St. Louis, MO, USA) for 4 h at 37 °C and 5% CO_2_. Following incubation, the culture media was gently aspirated and washed once with 1× PBS. NETosis was quantified by incubating the cells for 30 min at 37 °C, and 5% CO_2_ with SYTOX Green (S7020; Thermo Fisher Scientific, Waltham, MA, USA) diluted in PBS to a final concentration of 1 µM. Culture supernatant was gently aspirated, and 1× PBS was added carefully to each well. Fluorescence was quantified at excitation and emission wavelengths of 504 nm and 523 nm using a Cytation 5 Cell Imaging Multi-Mode Reader (BioTek, Santa Clara, CA, USA). To assess NET degradation, PBS was gently aspirated from each well, and NETs were incubated for 90 min (at 37 °C and 5% CO_2_) with normal control serum samples diluted to 5% in nuclease buffer. Each sample was tested in triplicate, with MNase- (10 U/mL) and DNase- (10 U/mL) treated wells serving as positive controls. Following incubation, the supernatant was gently discarded, and wells were washed once with 1× PBS. Fresh PBS was added to each well, and residual NETs were quantified by re-measuring SYTOX fluorescence at excitation and emission wavelengths of 504 nm and 523 nm using a Cytation 5 Cell Imaging Multi-Mode Reader (BioTek, Santa Clara, CA, USA). 

### 4.4. Immunofluorescence Microscopy of NETs

First, 1 × 10^5^ normal control neutrophils in serum-free RPMI media supplemented with L-glutamine were seeded onto 0.001% poly-L-lysine-coated coverslips. NET formation was induced by incubating neutrophils with 20 nM PMA for 4 h at 37 °C and 5% CO_2_. Following stimulation, culture media was gently aspirated, and cells were washed with 1× phosphate-buffered saline (PBS). To visualize the effects of NET degradation, each treatment was incubated with the cells for 90 min at 37 °C and 5% CO_2_. The treatments for visualization were 5% healthy control serum, MNase (10 U/mL) and DNase (10 U/mL) diluted in nuclease buffer (10 mM Tris–HCl pH 7.5, 10 mM MgCl_2_, 2 mM CaCl_2_, and 50 mM NaCl), or nuclease buffer as a control. Following incubation, culture media was gently aspirated and washed once with 1× PBS. Cells were fixed with 4% paraformaldehyde (PFA) at room temperature for 10 min, followed by overnight blocking in blocking buffer (10% fetal bovine serum (FBS) in PBS). For protein staining, fixed cells were incubated with polyclonal antibody to neutrophil elastase (Sigma-Aldrich, St. Louis, MO, USA) in a blocking buffer for 1 h at 4 °C, followed by FITC-conjugated secondary (Southern Biotech, Birmingham, AL, USA) for 1 h at 4 °C. Nuclear DNA was detected with Hoechst 33342 (H3570; Thermo Fisher Scientific, Waltham, MA, USA). Coverslips were mounted with Prolong Gold Antifade (Thermo Fisher Scientific, Waltham, MA, USA), and images were collected with a Cytation 5 Cell Imaging Multi-Mode Reader (BioTek, Santa Clara, CA, USA).

### 4.5. Immunofluorescence Microscopy of DHM Degradation by SLE Serum

To assess the degradation of DHMs coated on a 96-well microplate, 5% normal control or lupus serum diluted in nuclease buffer was added to the DHM-coated wells. For the SLE serum samples, the low-disease-activity serum was from a patient with SLEDAI-2K = 2. The high-disease-activity serum was from a patient with SLEDAI-2K = 16. The clinical activity level of SLE patients was calculated using SLEDAI-2K according to Gladman et al. [29]. DHMs were incubated for 12 h at 37 °C and 5% CO_2_. Following incubation, the culture supernatant was gently aspirated, and wells were washed once with 1× PBS. Degradation was assessed by incubating the wells with SYTOX Green diluted in PBS to a final concentration of 1 µM for 10 min at 37 °C and 5% CO_2_. Following incubation, the culture supernatant was gently aspirated, and wells were washed once with 1× PBS. Fresh PBS was added to each well, and images were captured by Cytation 5 Cell Imaging Multi-Mode Reader (BioTek, Santa Clara, CA, USA).

### 4.6. Statistical Analysis

The data were analyzed using GraphPad Prism 9.4.1 (458) (GraphPad Software, San Diego, CA, USA) and Excel 16.65 (Microsoft). Results are presented as mean ± SEM. Multiple groups were compared with the ANOVA test or its modification with post-hoc tests. Two groups were compared using the paired *t*-test. The differences between groups were reported as the following statistical values: * *p* < 0.05, ** *p* < 0.01, *** *p* < 0.001, and **** *p* < 0.0001.

## 5. Conclusions

We applied DHM chromatin mesh structures to visualize and quantify chromatin degradation in real time. Our results suggest that DHMs have different degradation mechanisms in response to nuclease versus serum degradation. DHM structures were more resistant to degradation when compared to NETs, yet DHMs were able to differentiate between varying chromatin degradation activity levels of SLE patient sera. These observations can guide the future development of DHMs in NET-related pathophysiological and diagnostic experiments.

## Figures and Tables

**Figure 1 ijms-24-03222-f001:**
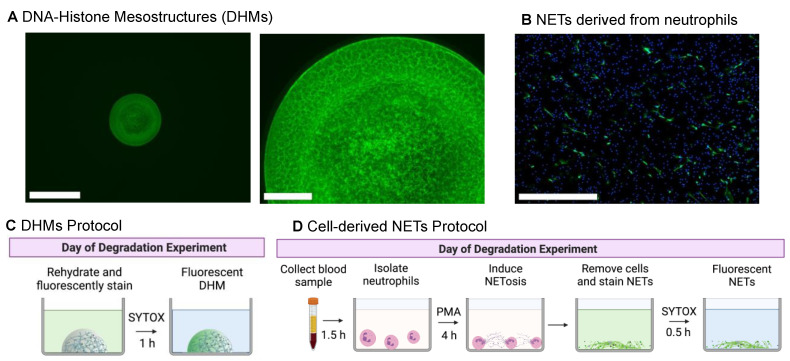
Initial structures of DNA–histone mesostructures (DHMs) and neutrophil extracellular traps (NETs) for degradation experiments. (**A**) One DHM micropatterned in each well of a 96-well microplate, single-well view (left) with a 650 μm scale bar, and magnified view of the fibrous structure (right), with a 125 μm scale bar. DHM structures were stained with SYTOX Green. (**B**) Fixed phorbol 12-myristate 13-acetate (PMA)-stimulated NETs as a visualization of the initial state before degradation experiments. NETs were stained with neutrophil elastase (NE; green), and neutrophils were detected with Hoechst 33342 (nuclear DNA, blue). We note that while NET visualization utilized NE, degradation experiments were quantified using SYTOX staining. The scale bar is 650 μm. A comparison of protocols needed to prepare DHMs (**C**) and NETs (**D**) for degradation experiments. Schematics created with BioRender.com, created on 1 October 2022.

**Figure 2 ijms-24-03222-f002:**
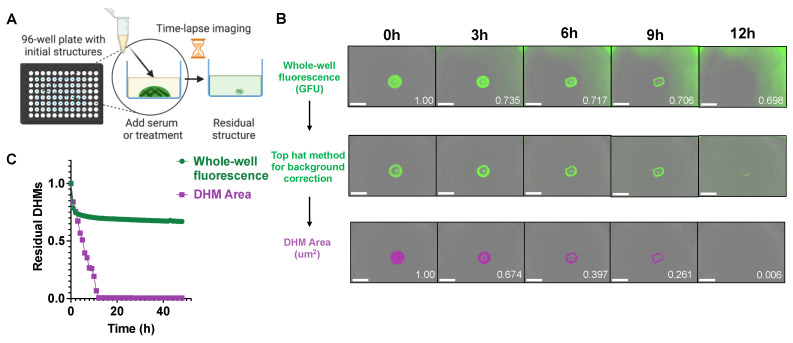
DHM degradation imaging procedure. (**A**) Each well of a 96-well microplate contained DHMs pre-stained with SYTOX Green and was imaged periodically by an in-incubator microscope. Schematic created with BioRender.com, created on 01 October 2022. (**B**) Time-lapse images of DHM degradation by 10 U/mL deoxyribonuclease I (DNase I) pre- and post-image processing. The collected green, fluorescent images can show signals from degraded DHM fragments in solution (top panel, normalized GFU values in white). Incucyte^®^ Software 2022A (Essen BioScience, Ann Arbor, MI, USA) image analysis was applied to segment the object area in each image (magenta traces, bottom panel, normalized area values in white). The scale bar is 800 μm. (**C**) Plots of DHM degradation by 10 U/mL DNase. The green fluorescence (green) and image-processed DHM area (magenta) are compared in the plot.

**Figure 3 ijms-24-03222-f003:**
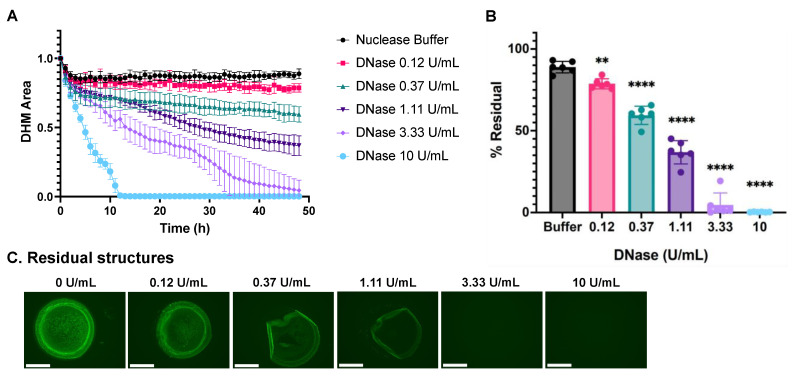
Dose-dependent degradation of DHMs by DNase I. DHM area over time during nuclease incubation is represented in kinetic curves (**A**) and end-point *%Residual* (**B**) plots. DHM residuals were compared with the control group, nuclease buffer, by one-way ANOVA, ** *p* < 0.01, and **** *p* < 0.0001. (**C**) Residual structures were visualized with fluorescent microscopy after 24 h incubation with a DNase concentration series. Structures were stained with SYTOX Green. The scale bar is 275 μm. All images are representative of *n* = 6 replicates per condition.

**Figure 4 ijms-24-03222-f004:**
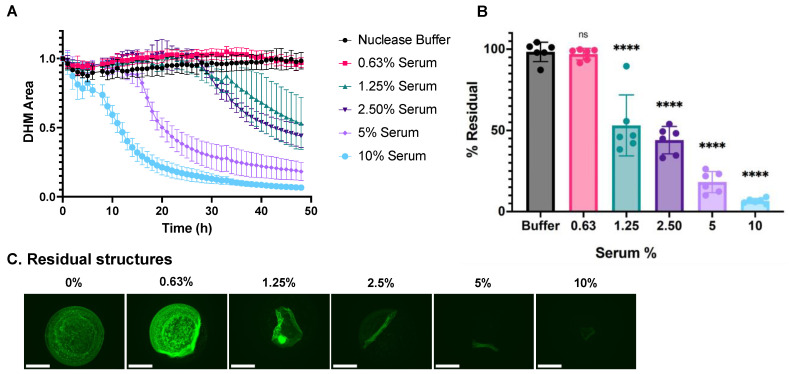
DHM degradation by normal human serum. (**A**) Kinetic curves of the dose-dependent DHM degradation by human serum. (**B**) End-point degradation of DHMs by serum comparing the final and initial amounts of DHMs. DHM residuals were compared with the control group, nuclease buffer, by one-way ANOVA, ns = not significant, and **** *p* < 0.0001. (**C**) Residual structures visualized with fluorescent microscopy after 24-h incubation with a serum concentration series. Structures were stained with SYTOX Green. The scale bar is 275 μm.

**Figure 5 ijms-24-03222-f005:**
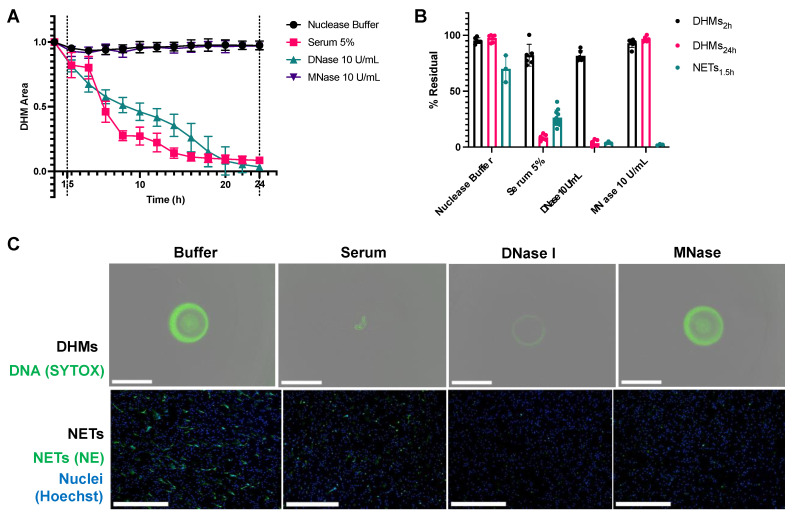
Differences are noted between cell-derived NETs and DHMs. (**A**) The kinetic curves of DHM degradation show that while NETs were degraded after a 1.5-h incubation, a time point marked with a dotted line, DHMs required a 24-h incubation for a comparable degradation extent. DHMs were degraded by serum and deoxyribonuclease I (DNase I), while buffer control and micrococcal nuclease (MNase) did not degrade DHMs. (**B**) DHMs degraded slowly when incubated with normal serum and nucleases relative to NETs. Residuals at time points for DHMs at 2 h showing minimal degradation, at 24 h showing the full extent of degradation, and for NETs, 1.5 h led to complete degradation. Readings for NET degradation were based on SYTOX Green signal. (**C**) Residual DHM structures were visualized with fluorescent microscopy after a 24-h incubation period with various degradation conditions (top panel) and NETs after a 1.5-h incubation period (bottom panel). DHM structures were stained with SYTOX Green. NETs were stained with neutrophil elastase (green), and neutrophils stained with Hoechst stain (blue) for visualization purposes. The scale bar is 650 μm.

**Figure 6 ijms-24-03222-f006:**
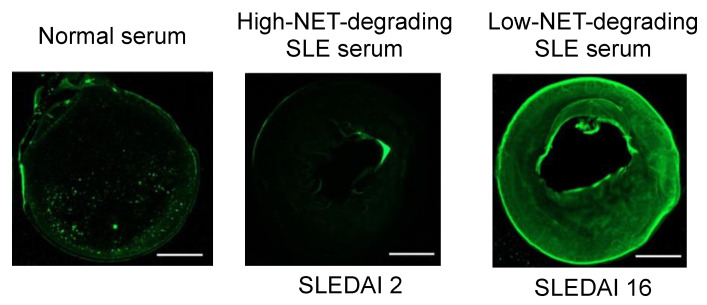
DHM degradation by normal serum and systemic lupus erythematosus (SLE) serum previously determined to have high- or low-NET-degrading capabilities. The high-NET-degrading SLE serum was from a patient with low disease activity (SLEDAI 2) while the low-NET-degrading SLE serum was from a patient with high disease activity (SLEDAI 16). DHMs pre-stained with 1 μM SYTOX Green were incubated with serum for 12 h. Note significant residual DHMs remaining after treatment with SLE serum from patient with high SLEDAI score. The images are representative of *n* = 3. The scale bar is 0.83 μm.

## Data Availability

The data presented in this study are available on request from the corresponding authors.

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
