# Peer review of "Visualization of Nuclease- and Serum-Mediated Chromatin Degradation with DNA–Histone Mesostructures"

_ijms, 2023, doi:10.3390/ijms24043222_

Round 1

Author Response

Dear Reviewer, 

Thank you, 

Reviewer 2 Report

My comments are mostly about clarifying the decisions you made during your research.

You wrote: “normal donor serum compared to serum from a lupus patient with high disease activity

[please describe how you chose this one patient, what were the key components of the serum relevant to your analysis, and how variable is the serum of such patients – and how would this affect your results?]

Here, we analyze 48 DHM degradation in more detail including comparison to NETs. The types of nuclease 49 tested was also expanded to include micrococcal nuclease (MNase), which has different 50 specificity from DNase I

[Please explain why you chose micrococcal nuclease. Which other nucleases would be appropriate and which did you also consider?]

Each DHM-containing microwell was monitored by time-lapse imaging using an in- 85 incubator microscope (Incucyte® S3, Sartorius) (Figure 2A). Figure 2B shows an example 86 of a DHM degradation time series upon incubation with 10 U/mL of DNase I. Under these 87 conditions, the raw green fluorescence units (GFU) readings captured both the DHM 88 structure remaining as well as the fluorescence of degradation products in the solution. 89 This resulted in a reading of 69.8% residual GFU at 12 hours despite the DHM structure 90 being completely degraded upon human visual inspection

[Which dataset or publications did you choose to identify the optimal time frame? For example, studies of penetration of COVID-19 spike proteins into human cells are seeking faster observation of the entire process]

commercially-available control male AB human serum (H4522; Sigma).

[which elements in the composition of this serum are most relevant to your study?]

NETs incubated with DNase I, MNase I, and serum conditions 165 degraded and had less green fluorescence than the nuclease buffer control condition

[please provide quantitative data in this text section]

Author Response

Dear Reviewer, 

Thank you

Round 2

Reviewer 1 Report

Authors have successfully addressed all the comments.